# Deregulated PP1α phosphatase activity towards MAPK activation is antagonized by a tumor suppressive failsafe mechanism

Ming Chen[1], Lixin Wan[2,3], Jiangwen Zhang[4], Jinfang Zhang[2], Lourdes Mendez[1], John G. Clohessy[1], Kelsey Berry[1], Joshua Victor[1], Qing Yin[3], Yuan Zhu[2], Wenyi Wei[2] & Pier Paolo Pandolfi [1]

The mitogen-activated protein kinase (MAPK) pathway is frequently aberrantly activated in advanced cancers, including metastatic prostate cancer (CaP). However, activating mutations or gene rearrangements among MAPK signaling components, such as Ras and Raf, are not always observed in cancers with hyperactivated MAPK. The mechanisms underlying MAPK activation in these cancers remain largely elusive. Here we discover that genomic amplification of the *PPP1CA* gene is highly enriched in metastatic human CaP. We further identify an S6K/PP1α/B-Raf signaling pathway leading to activation of MAPK signaling that is antagonized by the PML tumor suppressor. Mechanistically, we find that PP1α acts as a B-Raf activating phosphatase and that PML suppresses MAPK activation by sequestering PP1α into PML nuclear bodies, hence repressing S6K-dependent PP1α phosphorylation, 14-3-3 binding and cytoplasmic accumulation. Our findings therefore reveal a PP1α/PML molecular network that is genetically altered in human cancer towards aberrant MAPK activation, with important therapeutic implications.

[1] Cancer Research Institute, Beth Israel Deaconess Cancer Center, Department of Medicine and Pathology, Beth Israel Deaconess Medical Center, Harvard Medical School, Boston, MA 02215, USA. [2] Department of Pathology, Beth Israel Deaconess Medical Center, Harvard Medical School, Boston, MA 02115, USA. [3] Department of Molecular Oncology, H. Lee Moffitt Cancer Center and Research Institute, Tampa, FL 33612, USA. [4] School of Biological Sciences, The University of Hong Kong, Hong Kong 999077, China. Correspondence and requests for materials should be addressed to P.P.P. (email: ppandolf@bidmc.harvard.edu)

Activation of signaling pathways, such as the phosphoinositide-3-kinase (PI3K)/AKT and mitogen-activated protein kinase (MAPK), is regulated by feedback inhibition in both normal and cancer cells[1,2]. Evasion of feedback inhibition or failsafe mechanisms resulting from aberrant activation of major oncogenic pathways represent one of the critical mechanisms underlying tumor progression in tumors of diverse histological origin[3,4]. On the other hand, relief of negative feedback by anticancer drugs constitutes a major hurdle to limit the success of several targeted therapies[5]. Hence, identification of the key pathways that govern such regulation is of utmost importance for tumor-specific therapeutic targets.

Prostate cancer (CaP) is the most common malignancy found in men, and an estimated 1 in 7 men in the United States will be diagnosed with CaP during their lifetime[6]. In the past 25 years, CaP mortality has declined by nearly 40%; however, improvement in survival for patients with metastatic disease has not contributed substantially to the observed drop in CaP mortality[7]. More than 26,000 men in the United States die annually of metastatic CaP[6]. Recent whole-exome sequencing studies have revealed that copy number alterations, recurrent somatic mutations and genomic rearrangements are among the driving forces for metastatic castration-resistant prostate cancer (mCRPC) and have identified distinct molecular subtypes of mCRPC based on alterations in existing signaling pathways[8,9].

Co-activation of the PI3K/AKT and MAPK pathways is frequently observed in advanced and metastatic CaP and is found to be associated with disease progression and poor prognosis[10]. One of the dominant mechanisms underlying PI3K/AKT activation is inactivation of *PTEN* (*phosphatase and tensin homolog*)[11]. In contrast, the mechanisms underlying MAPK activation, to date, remain largely elusive since activating mutations or gene rearrangements among MAPK signaling components are rare in human CaP[8,12–17]. Notably, *PTEN* loss/PI3K–AKT activation occurs as an early event in the development of human CaP[18], leading to feedback inhibition on Ras/Raf/MAPK signaling[19,20] (Supplementary Fig. 1a-b). How human CaP evades this feedback inhibition to frequently co-activate the PI3K/AKT and MAPK signaling is also poorly understood. In view of these critical gaps in the field, we investigated the mechanistic basis of MAPK activation in metastatic human CaP.

Here we report an S6K/PP1α/B-Raf pathway that activates MAPK signaling in PI3K/AKT-driven cancers and is opposed by the promyelocytic leukemia (PML) tumor suppressor. We further demonstrate its importance in regulating CaP cell migration and invasion and in metastatic human CaP.

## Results

**Amplification of *PPP1CA* in metastatic human CaP.** It is now well recognized that the MAPK cascade is negatively regulated through inhibitory phosphorylation of components of the pathway, in particular, Raf kinases, the major upstream activators of MAPK signaling[21–25]. Raf kinases can not only be switched on by acquiring activating mutations, but also through phosphatase-mediated dephosphorylation at their inhibitory sites to relieve inhibition and to allow reactivation[21,22,26,27]. Given that activating mutations in Raf kinases are rare in human CaP, we postulated that aberrant phosphatase activity might promote Raf kinases activity and subsequent MAPK activation in metastatic human CaP.

We focused on PP2A and PP1, two major eukaryotic protein phosphatases that are reported to contribute to >90% of serine/threonine dephosphorylation and regulate a variety of cellular processes through the dephosphorylation of distinct substrates[28], and we initially sought to determine if genetic alterations to either

of these protein phosphatases could help establish a role in the context of metastatic cancer. Interestingly, the catalytic subunit of protein phosphatase 1α (PP1α), encoded by the *PPP1CA* gene in humans, is located on chromosomal band 11q13, one of the regions frequently amplified in comparative genomic hybridization (CGH) analysis of human CaP[29,30]. Moreover, enhanced cytoplasmic PP1α immunostaining strongly correlates with high Gleason Score[30], suggesting that PP1α may be involved in prostate tumorigenesis. To confirm the relevance of *PPP1CA* to human CaP, we evaluated the genomic status of *PPP1CA* in a recent array-based CGH (aCGH) data set[8] of 59 localized prostate cancer (LPC) and 35 mCRPC CaP. We found that *PPP1CA* was amplified in 7% of LPC and 17% of mCRPC, respectively (Fig. 1a). To independently validate the findings from the aCGH data set, we analyzed a recent large whole-exome sequencing data set[9] of 150 samples from mCRPC patients. Consistent with the aCGH analysis, *PPP1CA* was amplified in 25% of mCRPC (Fig. 1b). Notably, *PPP1CA*, which is ~2 Mb away from cyclin D1 a proto-oncogene also associated with metastatic CaP[18], was more frequently amplified than cyclin D1 in mCRPC (25% vs 5%, Fig. 1c). Also of interest, *PPP1CA* was co-amplified with cyclin D1 in 5 out of 7 cases where the latter was amplified (Fig. 1c). Thus, amplification of *PPP1CA* occurs frequently in metastatic human CaP.

**PP1α mediates *PML*-loss-induced MAPK activation.** We previously reported that co-deletion of *PTEN* and *PML* leads to MAPK reactivation in mouse prostate epithelial cells and human CaP cells and frequently occurs in metastatic human CaP[31] To determine if *PPP1CA* genomic amplification could cooperate with

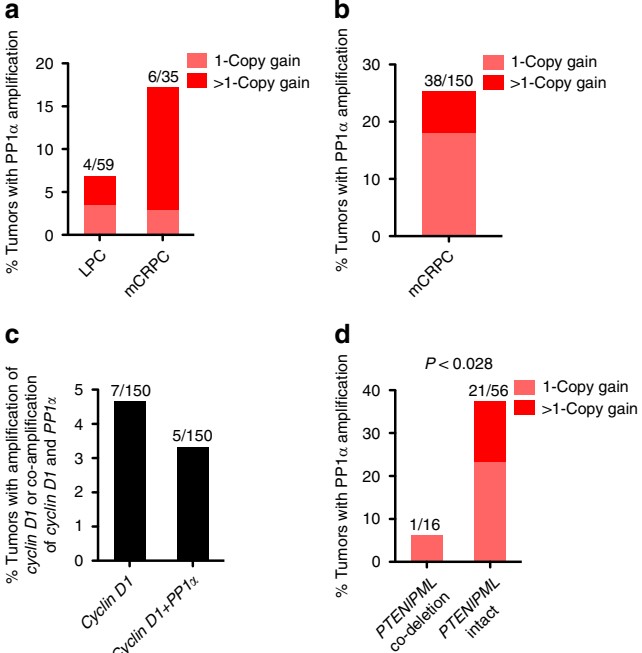

**Fig. 1** Amplification of *PPP1CA* occurs frequently in metastatic human CaP and is often mutually exclusive with co-deletion of *PTEN* and *PML*. **a**, **b** Bar graph showing the percentage of amplification of *PPP1CA* in LPC and mCRPC samples from the Grasso et al.[8] data set (**a**) and in mCRPC samples from the Robinson et al.[9] data set (**b**). **c** Bar graph showing the percentage of amplification of cyclin D1 alone or together with *PPP1CA* in mCRPC samples from the Robinson et al.[9] data set. **d** Association between genomic amplification of *PPP1CA* and co-deletion of *PTEN* and *PML* from the Robinson et al.[9] data set. Data were analyzed by Fisher's exact test, *P* < 0.05 was considered significant

co-deletion of *PTEN* and *PML* in metastatic human CaP, we determined whether amplification of *PPP1CA* was correlated with co-deletion of *PTEN* and *PML* in the Robinson et al.[9] data set. Strikingly, we found that *PPP1CA* genomic amplification and co-deletion of *PTEN* and *PML* were often mutually exclusive (Fig. 1d), supporting their proto-oncogenic functions in the same pathway. In keeping with this notion, overexpression of PP1α induced a marked increase in extracellular signal regulated kinase

(ERK) phosphorylation (Fig. 2a and Supplementary Fig. 2a). Conversely, knockdown of *PP1α* via small interfering RNA (siRNA) attenuated ERK phosphorylation in both LNCaP and PC3 cells (Fig. 2b), suggesting that PP1α is the principle phosphatase that positively regulates the MAPK signaling pathway. In contrast, although PP2A has previously been identified as the phosphatase mediating the dephosphorylation and reactivation of Raf kinases (Supplementary Fig. 2b)[22], the overexpression or

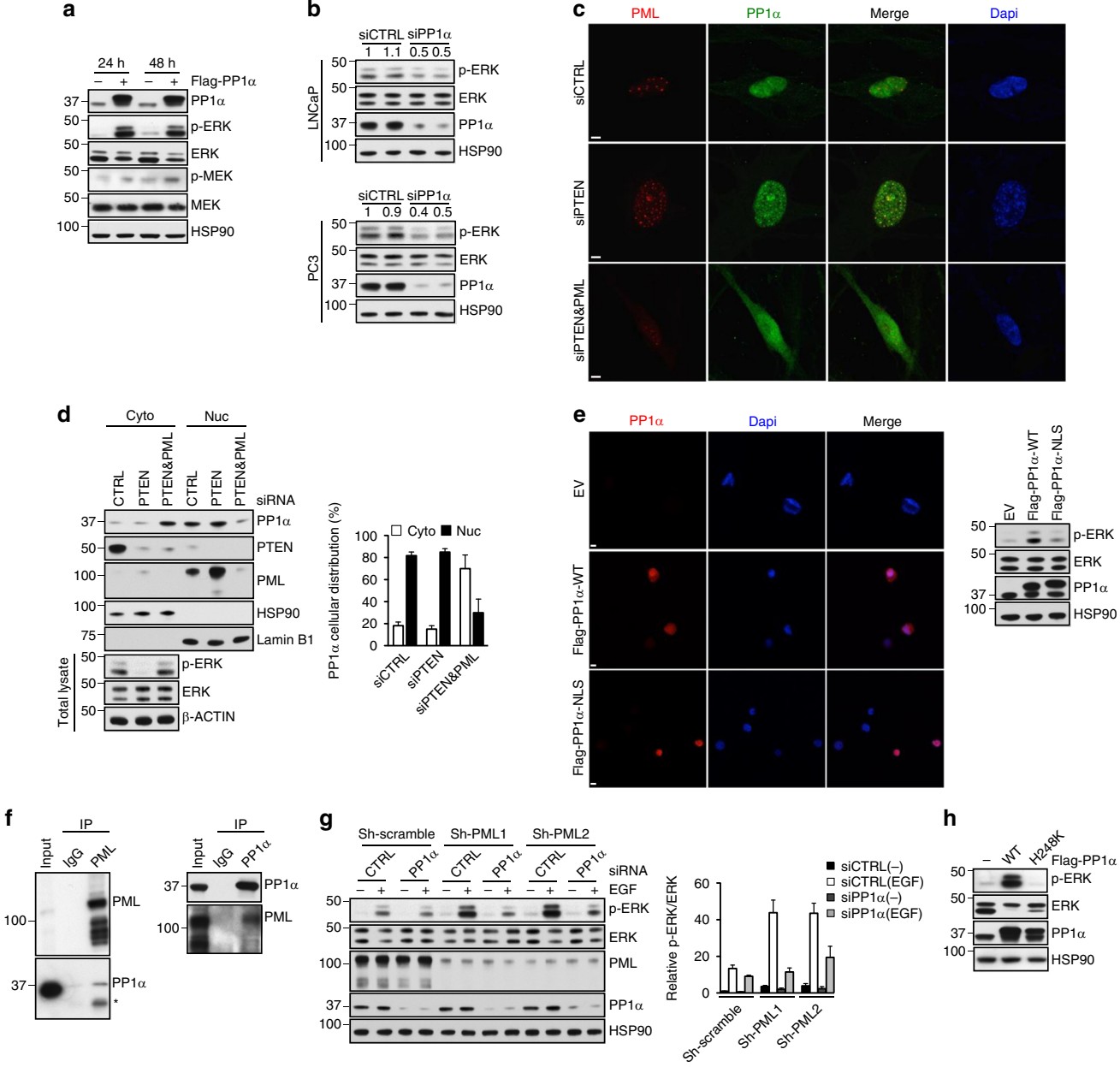

**Fig. 2** PP1α mediates *PML*-loss-induced MAPK activation. **a**, **b** Immunoblot (IB) analysis of lysates from 293T cells transfected with empty vector (EV) or Flag-PP1α for the indicated times (**a**), lysates from LNCaP or PC3 cells transfected with control or *PP1α* siRNA for 48 h (**b**). **c**, **d** Immunofluorescence (**c**) and fractionation (**d**) of PP1α protein in WI-38 cells transfected with control, *PTEN* or *PTEN* plus *PML* siRNA for 48 h. Quantification of the percentage of PP1α protein in the cytosolic and nuclear fraction was carried out with ImageJ software after it was normalized against cytosolic marker HSP90 and nuclear marker Lamin B1, respectively. Data shown are mean ± s.e.m. of three independent experiments. Scale bar, 5 μm. **e** Immunofluorescence and IB analysis of 293T cells transfected with EV, WT or NLS-tagged Flag-PP1α. Scale bar, 5 μm. **f** Endogenous reciprocal co-immunoprecipitation of PML and PP1α in PC3 cells. Input is 10% of total cell extracts used for immunoprecipitation. *Indicates a nonspecific band. **g** IB analysis of lysates from control or PML stable knocked-down PC3 cells transfected with control or *PP1α* siRNA for 48 h, followed by serum-starvation for 4 h and stimulation with 10 ng/ml EGF for 5 min. Quantification of p-ERK/ERK levels was carried out with ImageJ software. Numbers indicate the relative ratios to controls for phosphoprotein/total protein. Data shown are mean ± s.e.m. of three independent experiments. **h** IB analysis of lysates from 293T cells transfected with EV, WT or phosphatase-inactive (H248K) Flag-PP1α for 24 h

silencing of the PP2A catalytic subunit (PP2A-C) did not affect ERK phosphorylation in either 293T cells or CaP cell lines (Supplementary Fig. 2c-e), presumably due to its complex effects on various components of the MAPK cascade[32], and no consistent genomic alterations in *PPP2CA*, the gene encoding PP2A-C protein, have been found in mCRPC samples from the Robinson et al.[9] data set (Supplementary Fig. 2f).

Although co-deletion of *PTEN* and *PML* is often mutually exclusive with amplification of *PPP1CA,* we did not rule out the possibility that co-loss of *PTEN/PML* might lead to aberrant PP1α phosphatase activity, thereby contributing to *PML*-loss-induced MAPK activation. It is well established that PML is induced upon *PTEN* inactivation in a p53-dependent and -independent manner[4,33,34] and that PML, through its nuclear body (NB)-dependent scaffold activity, can interact with a variety of proteins, including phosphatases, to regulate their functions[35]. Additionally, PP1α is a known Rb phosphatase and, like PML, can bind Rb (Supplementary Fig. 2g) and cooperate with Rb function in the induction of cellular senescence[36]. We therefore considered that upon *PTEN* loss, PML might recruit PP1α into the PML-NBs to promote *PTEN*-loss-induced cellular senescence, a potent failsafe mechanism that restricts tumorigenesis[4], and simultaneously restrict PP1α-induced MAPK activation. WI-38 human diploid fibroblasts, the well-accepted cell model used to study the senescence with intact PTEN and PML protein expression, were used to test this possibility. We found that acute knockdown of *PTEN* in WI-38 cells induced senescence, PML upregulation, strong co-localization of PML and PP1α, and suppression of MAPK signaling (Supplementary Fig. 2h and Fig. 2c, d). On the other hand, we reasoned that in *PTEN/PML* double-null cells, due to the lack of sequestration to NBs, PP1α could contribute to MAPK activation through the dephosphorylation and reactivation of Raf kinases. In support of this hypothesis, the cytoplasmic accumulation of PP1α was significantly increased upon simultaneous knockdown of *PTEN* and *PML* along with concomitant MAPK reactivation in WI-38 cells (Fig. 2c, d). Moreover, compared to wild-type (WT) PP1α, a PP1α mutant constitutively targeted to the nucleus by fusing the NLS sequence derived from c-Myc[37] displayed a drastically decreased capacity to activate ERK (Fig. 2e), further corroborating the role of subcellular compartmentalization in the regulation of PP1α-induced ERK activation. Importantly, PML and PP1α interacted with each other in vivo in *PTEN*-null CaP cells. Immunoprecipitation of PML led to the co-immunoprecipitation of PP1α and vice versa in PC3 cells (Fig. 2f).

To further investigate whether PP1α might mediate *PML*-loss-driven MAPK activation, we used siRNA targeting *PP1α* in stable *PML* knockdown cells. We found that the induction of basal and EGF-stimulated ERK phosphorylation through stable knockdown of *PML* was suppressed by PP1α downregulation (Fig. 2g). Therefore, *PML*-loss-driven MAPK activation is mediated, at least in part, by PP1α. Notably, the phosphatase activity of PP1α is required for MAPK activation since expression of the phosphatase-inactive PP1α mutant (H248K)[38] did not affect ERK phosphorylation (Fig. 2h). In line with this, PC3 cells treated with tautomycin, a more selective inhibitor for PP1α[39–42], displayed a dose-dependent inhibition of EGF-induced ERK phosphorylation (Supplementary Fig. 2i). Thus, we conclude that in *PTEN* and *PML* double-null cells, sequestration of PP1α to NBs is impaired and in turn facilitates the aberrant cytosolic localization of PP1α and its subsequent activation of MAPK signaling.

**S6K induces PP1α phosphorylation, 14-3-3 binding and cytoplasmic accumulation**. To gain further mechanistic insights into how PP1α is delocalized into the cytoplasm, we examined which AGC kinase can trigger cytoplasmic accumulation of PP1α given that the AGC family kinases are commonly activated upon *PTEN* loss[43]. We found that S6K1, but not other AGC kinases such as AKT and SGK, phosphorylated PP1α in vivo (Fig. 3a), induced the interaction of PP1α with 14-3-3γ (Fig. 3b) and led to increased cytoplasmic accumulation of PP1α (Fig. 3c, d). Notably, knockdown of *S6K1* via siRNA largely suppressed PP1α-induced ERK activation (Fig. 3e), suggesting that activation of ERK by PP1α is dependent on S6K. In support of PP1α being an S6K substrate, we identified two highly conserved imperfect AGC family kinase-recognition motifs (RxRxxpS/T) located at S224/T226 and T320 of PP1α protein, respectively (Fig. 3f). We generated PP1α mutants in which S224/T226 (S224A/T226A), T320 (T320A) or all three sites (3A) were mutated to alanine. In vitro kinase assays confirmed that recombinant S6K1 could also phosphorylate PP1α (Fig. 3g). Moreover, S224A/T226A and 3A PP1α mutants, but not T320A, displayed drastically reduced S6K1-dependent PP1α phosphorylation (Fig. 3g). Consequently, S224A/T226A and 3A PP1α mutants had a lower capacity to interact with 14-3-3 and to activate MAPK signaling (Fig. 3h, i), suggesting that S6K-mediated PP1α phosphorylation on S224/T226 is critical for the binding of PP1α with 14-3-3 and for the ability of PP1α to activate MAPK.

**PP1α dephosphorylates B-Raf inhibitory phosphorylation sites**. Next, we asked if PP1α could interact with Raf family kinases. Immunoprecipitation of PP1α revealed a strong and specific interaction between PP1α and B-Raf and to a lesser extent, A-Raf, while PP1α did not interact with C-Raf or other components of the MAPK cascade, such as MEK and ERK (Fig. 4a, the left panel). Given that A-Raf shows weak interaction with PP1α and has low intrinsic activity towards MAPK activation[44], further studies investigating the effect of PP1α on Raf family kinases were focused on B-Raf. The interaction between PP1α and B-Raf was further confirmed by reciprocal immunoprecipitation in which PP1α co-immunoprecipitated with anti-B-Raf precipitates (Fig. 4a, the right panel).

The observation that PP1α and B-Raf interact and that the phosphatase activity of PP1α is required for MAPK activation led us to investigate whether B-Raf might be a putative substrate of PP1α. We therefore asked if B-Raf itself is dephosphorylated by PP1α. Since ERK-mediated feedback inhibitory phosphorylation on S151, T401, S750 and T753 of B-Raf protein negatively regulates its kinase activity[22], we reasoned that PP1α could dephosphorylate these inhibitory sites and, in turn, relieve feedback inhibition and reactivate MAPK. To address whether PP1α activates B-Raf through these inhibitory sites, we made use of B-Raf protein mutants in which an individual inhibitory site, as well as all four sites (4A), were mutated to alanine. As expected, cells overexpressing either PP1α or WT B-Raf had higher ERK phosphorylation than cells transfected with empty vector (Fig. 4b). Additionally, cells overexpressing both PP1α and WT B-Raf displayed even higher ERK phosphorylation (Fig. 4b), suggesting that PP1α can enhance B-Raf activity. On the other hand, the 4A B-Raf mutant, but not single alanine B-Raf mutants, displayed an enhanced basal ability to activate MAPK compared to WT B-Raf (Fig. 4b), indicating that these four sites indeed negatively regulate B-Raf activity, while dephosphorylation of a single inhibitory site is not sufficient to increase B-Raf activity. Critically, the 4A B-Raf mutant was insensitive to PP1α activation (Fig. 4b). In contrast, mutation of other known B-Raf inhibitory phosphorylation sites, including S365A/S429A/T440A[24], S465A/S467A[25] and S614A[23], to alanine failed to blunt PP1α-mediated ERK activation (Supplementary Fig. 2j-l). Therefore, PP1α appears to exert its effect on B-Raf primarily through the ERK-regulated inhibitory sites.

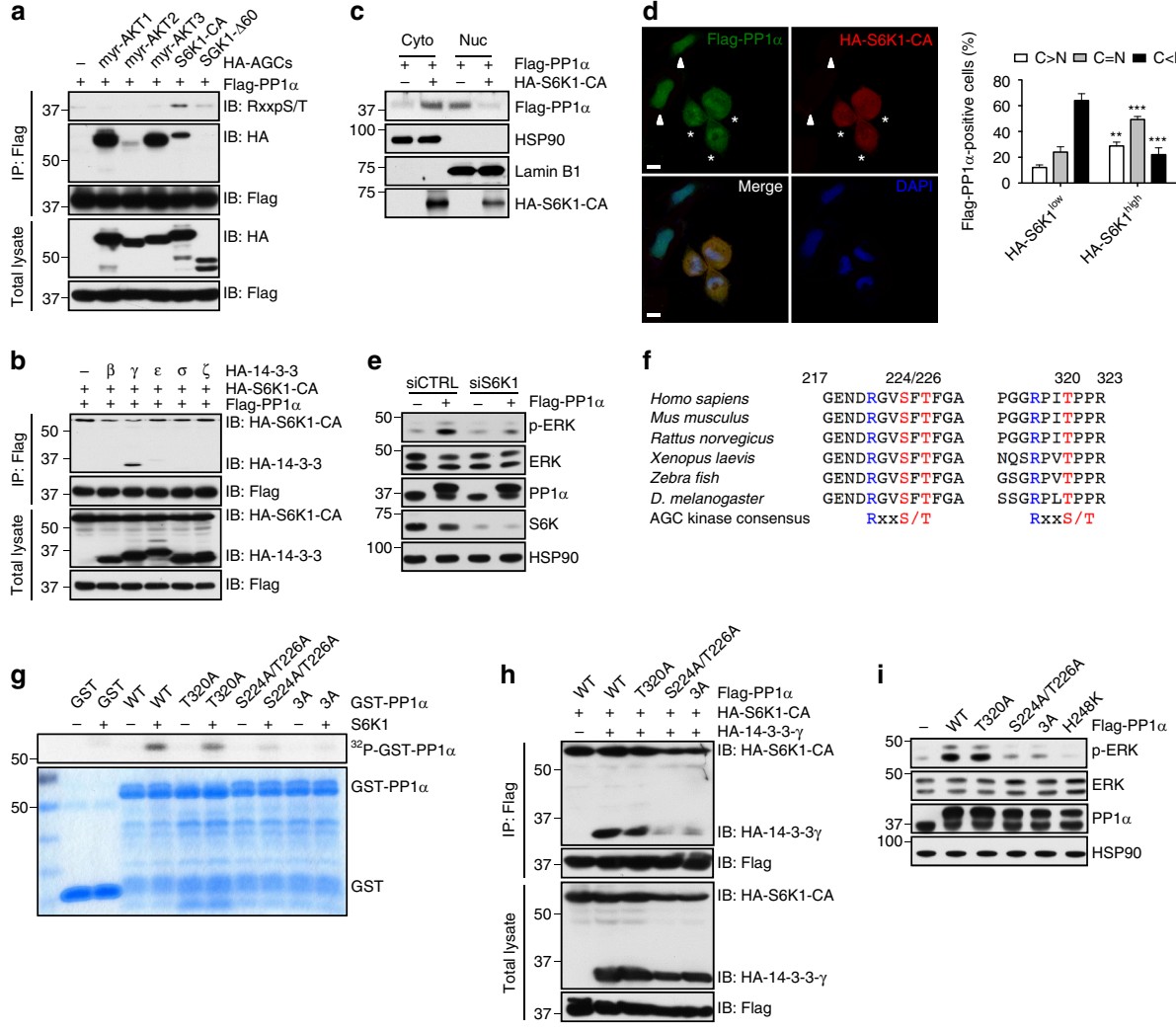

**Fig. 3** S6K1 phosphorylates PP1α, induces the binding of PP1α with 14-3-3γ and triggers its cytoplasmic accumulation. **a, b** IB analysis of total lysates and immunoprecipitates from 293T cells transfected with Flag-PP1α plus EV or the indicated HA-tagged constitutively active (CA) AGC family kinases for 48 h (**a**), from 293T cells transfected with Flag-PP1α, HA-S6K1-CA plus EV or the indicated HA-tagged 14-3-3 isoforms for 48 h (**b**). **c** Fractionation of 293T cells transfected with Flag-PP1α plus EV or HA-S6K1-CA for 48 h. **d** Immunofluorescence and quantitation of 293T cells transfected with Flag-PP1α and HA-S6K1-CA for 48 h. Arrowhead, HA-S6K1$^{low}$ cells; asterisk, HA-S6K1$^{high}$ cells. Data shown are mean ± s.e.m. of three independent experiments. **$P < 0.01, ***P < 0.001$ by unpaired two-tailed $t$-test. Scale bar, 10 μm. **e** IB analysis of lysates from 293T cells transfected with control or *S6K1* siRNA in the absence and in the presence of Flag-PP1α for 24 h. **f** A schematic showing the two highly conserved putative S6K sites, S224/T226 and T320, in PP1α protein. **g** In vitro S6K-mediated PP1α kinase assays. Bacterial-expressed WT or mutant GST-PP1α was purified and incubated with S6K1 in the kinase buffer with [γ-$^{32}$P] ATP. Reaction was stopped by sample buffer and resolved by SDS-PAGE. **h, i** IB analysis of total lysates and immunoprecipitates from 293T cells transfected with the indicated WT or mutant Flag-PP1α constructs, HA-S6K1-CA plus EV or HA-14-3-3γ for 48 h (**h**), lysates from 293T cells transfected with EV or the indicated WT or mutant Flag-PP1α constructs for 24 h (**i**).

To determine which site could be dephosphorylated by PP1α, we purified GST-3A-B-Raf proteins in which one inhibitory site was WT and the other three sites were mutated to alanine. GST-WT-B-Raf and GST-4A-B-Raf protein were included as the positive and negative control, respectively. We then phosphorylated GST-B-Raf in vitro by incubating B-Raf with recombinant ERK2 protein, and next used the phosphorylated form of B-Raf as a substrate for recombinant PP1α. We confirmed that GST-B-Raf was phosphorylated by ERK2 in vitro[22], mainly on S151 and T753 (Fig. 4c), and found that PP1α dephosphorylated B-Raf on both ERK phosphorylation sites (Fig. 4d). Thus, PP1α appears to dephosphorylate these inhibitory sites of B-Raf, triggering relief of feedback inhibition and consequent activation of the MAPK pathway.

**PP1α promotes CaP cell invasiveness via MAPK signaling**. The identification of *PPP1CA* amplification in metastatic human CaP is consistent with a pro-metastatic role for *PPP1CA* in the prostate (Fig. 1a, b). We therefore examined whether overexpression of PP1α affects CaP cell migration and invasion. Indeed, we found that LNCaP cells stably overexpressing PP1α exhibited higher ERK activation along with significantly increased cell migration and invasion (Fig. 4e). Notably, treatment with the MEK inhibitor, U0126, in LNCaP cells repressed not only basal but also PP1α-induced cell migration and invasion (Fig. 4e). Similar results were also obtained in PC3 cells (Supplementary Fig. 2m). These functional data, together with the human genetic and mechanistic analyses, implicate *PPP1CA* as a pro-metastatic proto-oncogene in human CaP and MAPK signaling as one of the key downstream effectors of PP1α-induced cell invasiveness.

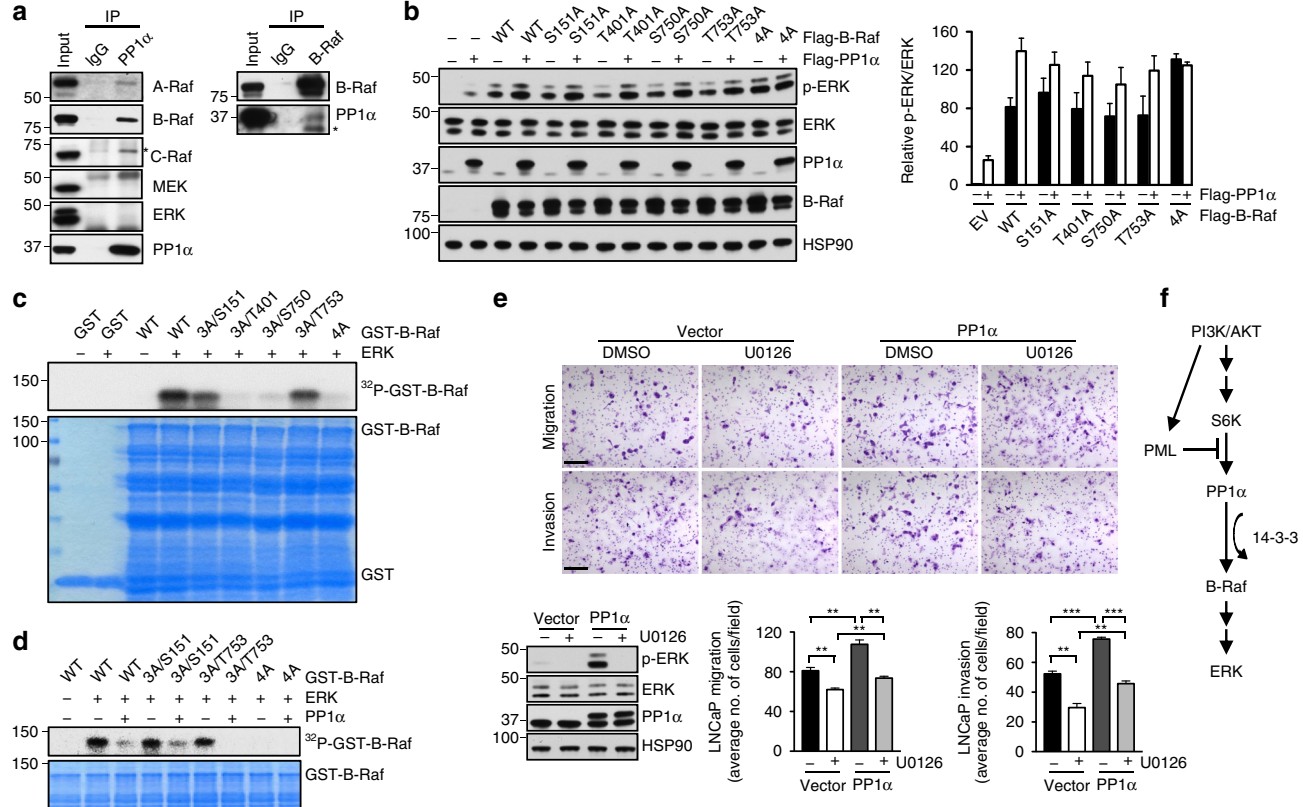

**Fig. 4** PP1α acts as a B-Raf activating phosphatase and promotes CaP cell invasiveness through activation of MAPK signaling. **a** Endogenous co-immunoprecipitation of PP1α with A-Raf, B-Raf, C-Raf, MEK and ERK (the left panel) or B-Raf with PP1α (the right panel) in PC3 cells. Input is 10% of total cell extracts used for immunoprecipitation. *Indicates a nonspecific band. **b** IB analysis of lysates from 293T cells transfected with the indicated WT or mutant Flag-B-Raf constructs plus EV or Flag-PP1α for 24 h. Quantification of p-ERK/ERK levels was carried out with ImageJ software. Numbers indicate the relative ratios to controls for phosphoprotein/total protein. Data shown are mean ± s.e.m. of three independent experiments. **c** In vitro ERK-mediated B-Raf kinase assays. Bacterial-expressed WT or mutant GST-B-Raf was purified and incubated with ERK2 in the kinase buffer with [γ-$^{32}$P] ATP. Reaction was stopped by sample buffer and resolved by SDS-page. **d** In vitro ERK-mediated B-Raf kinase and PP1α phosphatase assay. Bacterial-expressed WT or mutant GST-B-Raf was purified and incubated with ERK2 in the kinase buffer with [γ-$^{32}$P] ATP in the absence or presence of recombinant PP1α. Reaction was stopped by sample buffer and resolved by SDS-PAGE. **e** Representative images and quantitation of migrated and invaded LNCaP cells in the migration and invasion assay. LNCaP stable cells were subjected to migration (24 h) or invasion assay (48 h) in the absence or presence of 20 μM U0126. IB analysis confirmed the expression of phosphor-ERK and PP1α. Data shown are mean ± s.e.m. of three independent experiments. **P < 0.01, ***P < 0.001 by unpaired two-tailed t-test. Scale bar, 100 μm. **f** A model of the regulation of MAPK activation in *PTEN*-null cells by S6K, PP1α, B-Raf and PML

## Discussion

Integrated genetic and molecular analyses allowed us to identify an S6K/PP1α/B-Raf pathway towards the aberrant activation of MAPK signaling. We find that this pathway is suppressed by the PML tumor suppressor through sequestration of PP1α into NBs in *PTEN*-null cells as a result of failsafe mechanisms evoked by *PTEN* loss (Fig. 4f). Although the critical role of PP1- and PP2A-mediated dephosphorylation in the activation of Raf kinases has previously been reported[26,27,45,46], we show here for the first time that (1) PP1α is amplified in metastatic human CaP and is the principle phosphatase positively regulating MAPK activation and that (2) S6K-mediated PP1α cytoplasmic accumulation is essential for the activation of MAPK by PP1α. Importantly, given that PML is frequently lost in human cancer[47], our study suggests that aberrant cytoplasmic retention of PP1α caused by *PML* loss or the amplification of *PPP1CA* might represent a common mechanism underlying MAPK activation in cancers that lack activating mutations or gene rearrangements among MAPK signaling components, such as breast cancer and CaP[13–17,48,49]. Notably, since nuclear PP1α has been shown to activate Rb tumor suppressor through dephosphorylation[50], PML and PML-NBs do serve as rheostats to switch PP1α activity from tumor suppressive to tumor promoting. It is also worth noting that although both

PML and 14-3-3γ interact with PP1α, neither of them affect PP1α phosphatase activity towards dephosphorylating its respective nuclear and cytosol substrates (Supplementary Fig. 2n,o), suggesting that PML or 14-3-3γ primarily functions as a scaffold/chaperone for PP1α rather than as a direct regulator of PP1α phosphatase activity.

Additionally, as S6K is a downstream target of the ERK pathway[51,52], our study suggests that in the context of *PML* loss or *PPP1CA* amplification, the S6K/PP1α/B-Raf/ERK pathway represents a feed-forward loop supporting sustained ERK activation. However, we and others have previously also shown that AKT/mTOR/S6K activation triggers a negative feedback on MAPK signaling pathway, presumably as a result of the upstream IRS inactivation induced by S6K[19,53] (Supplementary Fig. 1). Thus, depending on the genetic context, S6K is a double-edge sword in the regulation of ERK activation, since it can act as both a suppressor of ERK activation, in the context of intact PML function, and as a promoter of sustained ERK activation, in the context of *PML* loss or *PPP1CA* amplification.

Collectively, our findings have important implications for tumorigenesis at large because simultaneous activation of MAPK and PI3K/AKT signaling undoubtedly represents a devastating force in propelling cancers into more advanced and aggressive

diseases. Since MAPK activity can be suppressed/inhibited by small pharmacological inhibitors, this study provides a compelling rationale for investigating whether patients with co-deletion of *PTEN* and *PML* or amplification of *PPP1CA* may benefit significantly from combinatorial therapy targeting both AKT/mTOR and MAPK signaling that is worthy of further investigations.

## Methods

**Plasmids, reagents and antibodies.** Human WT and c-terminal NLS[c-Myc]-tagged[37] PP1α complementary DNA (cDNA) were cloned into pCMV-Tag2B vector to generate PP1α expression plasmid. pCDNA3.1-hygro-B-Raf was purchased from Addgene. B-Raf and PML-I cDNA were subcloned into pGEX-4T-1 vector and used to express GST-B-Raf and GST-PML protein. All mutant constructs of B-Raf and PP1α were generated using a QuickChange Lightning Site-Direct Mutagenesis (Agilent Technologies) and all mutations were confirmed by sequencing. HA-Myr-AKT1/AKT2/AKT3, HA-SGK1Δ60, HA-S6K1-CA and HA-14-3-3 isoforms have been previously described[54]. The SMART pool or two independent siRNA duplexes targeted to PML, PTEN, PP1α, PP2A-C, S6K1 and control non-target siRNA were purchased from Dharmacon or Sigma. The sequences for the siRNA are listed in Supplementary Table 1. The target sequences in the pLKO-PML shRNA vector against human PML were 5′-GTGTACGCCTTCTCCATCAAA-3′ and 5′-CACCCGCAAGACCAACAACAT-3′. Tautomycin was from Enzo Life Sciences. U0126 was from Selleck Chemicals. EGF, Lipofectamine 2000, Lipofectamine RNAiMAX, RPMI, Dulbecco's modified Eagle's medium (DMEM), Opti-MEM reduced serum media and fetal bovine serum (FBS) were from Invitrogen. Polybrene and puromycin were from Sigma. We used the following primary antibodies for immunoblotting: anti-p-ERK (Cell Signaling Technology, 9101, 1:1000), anti-ERK (Cell Signaling Technology, 9102, 1:1000), anti-p-MEK (Cell Signaling Technology, 9154, 1:1000), anti-MEK (Cell Signaling Technology, 9126, 1:1000), anti-A-Raf (Cell Signaling Technology, 4432, 1:1000), anti-B-Raf (Santa Cruz Biotechnology, sc-5284, 1:1000), anti-B-Raf (Santa Cruz Biotechnology, sc-9002, 1:1000), anti-C-Raf (Cell Signaling Technology, 9422, 1:1000), anti-p-AKT substrate (RXXS*/T*) (Cell Signaling Technology, 9614, 1:1000), anti-p-AKT (Cell Signaling Technology, 9271, 1:1000), anti-AKT (Cell Signaling Technology, 9272, 1:1000), anti-p-S6K (Cell Signaling Technology, 9234, 1:1000), anti-S6K (Cell Signaling Technology, 2708, 1:1000), anti-IRS1 (Cell Signaling Technology, 3407, 1:1000), anti-IRS2 (Cell Signaling Technology, 4502, 1:1000), anti-PP2A-C (Cell Signaling Technology, 2259, 1:1000), anti-p-CREB (Cell Signaling Technology, 9198, 1:1000), anti-GAPDH (Cell Signaling Technology, 2118, 1:6000), anti-PML (Bethyl Laboratories, A301-167A, 1:2000), anti-HA (Santa Cruz Biotechnology, sc-805, 1:1000), anti-HSP90 (BD Biosciences, 610419, 1:4000, anti-PP1α (Novus Biologicals, NB-110-57428, 1:1000), anti-Flag (Sigma-Aldrich, F1804, 1:4000), anti-β-actin (Sigma-Aldrich, A5316, 1:4000), anti-Rb (Santa Cruz Biotechnology, sc-50, 1:1000) and anti-Lamin B1 (Abcam, ab16048, 1:1000).

**Cell culture, transfection and lentivirus production.** All cell lines were purchased from the American Type Culture Collection, cultured in RPMI or DMEM supplemented with 10% FBS and tested for mycoplasma contamination every month. Transfections were performed using Lipofectamine 2000 or Lipofectamine RNAi-MAX reagent (Invitrogen) according to the manufacturer's instruction. In brief, $1 \times 10^5$ cells were transfected with 1 μg of DNA plasmids or 50 nM siRNA in a 6-well dish. Cells were recovered into completed media after 12 h of transfection and then harvested at the indicated times. For lentivirus production, human PP1α cDNA was subcloned into the pWPI-Puro lentiviral vector to generate pWPI-Puro-PP1α. pWPI-Puro Vector or pWPI-Puro-PP1α (6 μg), pMD2.G (1.5 μg) and psPAX2 (4.5 μg) were co-transfected into 293T cells using Lipofectamine 2000. The virus supernatants were collected 48 h after transfection and filtered through a 0.45 μm filter. Freshly made virus supernatants supplemented with 4 μg/ml polybrene were added to exponentially growing LNCaP and PC3 cells. After 5 h, fresh medium was added. CaP cells were then selected with 2 μg/ml puromycin (Sigma) for 48 h after 2-day infection and then used for the cell migration and invasion assay.

**Immunofluorescence.** Cells were grown on coverslips, fixed with 4% paraformaldehyde and permeabilized with ice-cold methanol. Cells were rinsed with phosphate-buffered saline, blocked with 10% goat serum and then incubated with primary antibody overnight, followed by incubation with Alexa Fluor-conjugated secondary antibodies (Life Technologies). Coverslips were mounted with ProLong Gold Antifade reagent with DAPI (Life Technologies). The following primary antibodies were used for immunofluorescence: anti-Flag (Sigma-Aldrich, F1804, 1:400), anti-HA (Santa Cruz Biotechnology, sc-805, 1:400), anti-PML (Santa Cruz Biotechnology, sc-966, 1:400) and anti-PP1α (Bethyl Laboratories, A300-904A, 1:400). The stained slides were visualized by a bright-field or confocal microscope. Senescence-associated β-galactosidase activity in prostate tissue was measured with the senescence detection kit (Calbiochem) on 5 μm thickness frozen section.

**Western blotting and immunoprecipitation.** Cell lysates or prostate tissues were prepared in RIPA buffer (Sigma) supplemented with protease (Roche) and phosphatase (Sigma) inhibitor. Proteins were separated on 4–12% Bis-Tris gradient gels (Invitrogen), transferred to polyvinylidine difluoride membranes (Immobilon P, Millipore) and the blots were probed with the indicated antibodies. Densitometry quantification was performed with ImageJ. Nuclear/cytoplasmic fractionation was performed as previously described[55]. For immunoprecipitation, cells were lysed in lysis buffer (50 mM Tris at pH 7.5, 10% glycerol, 5 mM MgCl$_2$, 150 mM NaCl, 0.2% NP-40, protease (Roche) and phosphatase (Sigma) inhibitor) and the lysates were incubated with anti-PML (Santa Cruz Biotechnology, sc-966, 1:100) or anti-PP1α (Invitrogen, 43-8100, 1:50) or anti-B-Raf (Santa Cruz Biotechnology, sc-5284, 1:100) or anti-Rb (Santa Cruz Biotechnology, sc-102, 1:100) or anti-Flag (Sigma-Aldrich, F1804, 1:100) antibody overnight at 4 °C. The protein G sepharose (GE Healthcare) was then added and incubated for another 2 h. The immunoprecipitates were washed with wash buffer (50 mM Tris at pH 7.5, 5 mM MgCl$_2$, 150 mM NaCl, 0.1% NP-40, protease (Roche) and phosphatase (Sigma) inhibitor) three times and eluted with 2× SDS sample buffer. Prostate tissues from WT and Pten[pc][−/−] mice as well as Primary Pten[lox/lox] MEFs and their transduction with retrovirus-expressing PURO-IRES-GFP or Cre-PURO-IRES-GFP have been previously described[4]. Uncropped scans for western blotting presented in main figures are provided in Supplementary Figure 3.

**In vitro kinase and phosphatase assays.** In vitro kinase assays were performed as previously described[56]. Briefly, bacterial-expressed GST-PP1α-H248 (the phosphatase-dead mutant to avoid the autodephosphorylation of PP1α) or GST-B-Raf was purified with Glutathione Sepharose 4B (GE Healthcare) according to the manufacturer's instructions. To determine the residue(s) on PP1α protein phosphorylated by S6K1, recombinant S6K1 (R&D Systems) was incubated with 1 μg of GST-PP1α-H248. For the in vitro phosphatase assays, recombinant ERK2 (R&D Systems) was incubated with 1 μg of GST-B-Raf in the absence or presence of PP1α (Lifespan Bioscience) or PP2A-C (Abcam). Both reactions were incubated in kinase assay buffer (50 mM Tris-HCl pH 7.5, 2 mM MgCl$_2$, 0.1 mM EDTA, 2 mM DTT, 0.1 mM ATP) with 5 μCi [γ-$^{32}$P] ATP. For PP1α treatment, 1 mM MnCl$_2$ was added to the kinase assay buffer to promote PP1α activity. The reaction was initiated by the addition of GST-PP1α-H248 or GST-B-Raf in a volume of 30 μl for 30 min at 30 °C followed by the addition of sodium dodecyl sulfate-polyacrylamide gel electrophoresis (SDS-PAGE) sample buffer to stop the reaction before resolved by SDS-PAGE. To determine B-Raf kinase activity towards phosphorylating MEK1, in vitro kinase assays were performed as described previously[57]. Briefly, B-Raf kinase was immune-purified from 293T cells transfected with Flag-B-Raf constructs. GST-MEK1 was expressed in BL21 *Escherichia coli* and purified using Glutathione Sepharose 4B media (GE Healthcare). B-Raf kinase was incubated with 0.2 μg of GST-MEK1 in the absence or presence of PP1α or PP2A-C in kinase assay buffer (10 mM HEPES pH 8.0, 10 mM MgCl$_2$, 1 mM dithiothreitol, 0.1 mM ATP). Reaction was initiated by the addition of GST-MEK1 in a volume of 30 μl for 15 min at 30 °C followed by the addition of SDS-PAGE sample buffer to stop the reaction before resolved by SDS-PAGE. Nuclear PP1α phosphatase activity was determined by in vitro phosphatase assays using CREB as the substrate as described previously[58]. Phosphorylated CREB protein was immune-purified from 293T cells transfected with Flag-CREB. The phosphatase assay was carried out using Flag-CREB and PP1α in 1X NEBuffer for PMP supplemented with 1 mM MnCl$_2$ (New England Biolabs). Bacterially expressed and purified GST-PML was added as indicated in the experiments.

**Cell migration and invasion assay.** CaP cells stably expressing pWPI-Vector or pWPI-PP1α were detached into single-cell suspension. LNCaP ($1 \times 10^5$) or PC3 ($5 \times 10^3$ and $5 \times 10^4$ for migration and invasion assay, respectively) cells in 100 μl of 0.1% FBS-containing RPMI medium in the absence or presence of 20 μM U0126 were placed into the top chamber of 8 μm transwell inserts for migration assay or Matrigel-coated transwell inserts for invasion assay (BD Biosciences). A volume of 600 μl of 10% FBS-containing RPMI in the absence or presence of 20 μM U0126 were added to the bottom wells. After 24 h or 48 h (for LNCaP invasion assay only), cells on the upper surface of the inserts were removed with a cotton swab. Migrated cells were fixed in 10% formalin, then stained with 0.2% crystal violet for 10 min. Cells were counted in four microscopic fields under ×20 magnification. Results are representative of three independent experiments.

**Array CGH analysis.** The data were downloaded from GEO database (Grasso: GSE35988) or cBioportal (for Robinson et al.[9] data set) with focus on the aCGH data sets. The R scripts were used to process the data. The cutoff threshold we used is −0.35 to −0.8 as heterozygous deletions, those lower than −0.8 as homozygous deletions, 0.6 to 0.8 as 1-copy gain and those higher than 0.8 as 1-copy gain.

**Statistical analysis.** The 2 × 2 contingency tables were constructed and were used to analyze categorical data (for example, copy number alteration). The data sets were compared using Fisher's exact test. For analysis of average data, data sets were compared using unpaired two-tailed Student's *t*-tests. *P*-values of <0.05 were considered to be statistically significant.

**Data availability.** The authors declare that the data supporting the findings of this study are available within the article and its Supplementary Information files. All relevant data are available from the authors upon request.

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

## Acknowledgements

We thank all the members of the Pandolfi lab for critical comments, and Lauren Southwood and Elizabeth Stack for editing the manuscript. We thank the BIDMC Confocal Imaging Core facility. M.C. was supported in part by the DOD Prostate Cancer

Research Program (PCRP) Postdoctoral Training Award. This work was supported by NIH grants R01CA-142780, R01CA-142874 and R35CA-197529 to P.P.P.

## Author contributions

M.C., L.W., J.F.Z., K.B., J.V., Q.Y. and Y.Z. performed the experiments. M.C., L.W., W. W. and P.P.P. conceived and designed the experiments. W.W. and P.P.P. supervised the study. J.W.Z. performed all bioinformatic analyses. M.C., L.W., J.W.Z., W.W. and P.P.P. analyzed the data. M.C., L.W., L.M., J.G.C., W.W. and P.P.P. wrote the manuscript. All authors critically discussed the results and the manuscript.

## Additional information

**Competing interests:** The authors declare no competing financial interests.

