## [Peer Review File · Nature Communications]

Reviewers' comments:

Reviewer #1 (Remarks to the Author):

In this paper the authors report the positive regulation of ERK signalling by PP1a. Investigating the mechanism they find that PP1a is sequestered in nuclear PML bodies, from where it can be released by S6K mediated phosphorylation and binding to 14-3-3g. It then can translocate to the cytosol to activate BRAF by dephosphorylation of inhibitory residues. This is an interesting finding. Although the regulation of RAF kinases by PP1 and PP2 has been reported before (Abraham et al., 2000; Dhillon et al., 2002; Jaumot and Hancock, 2001; Strack, 2002), the aspect of the spatial regulation is new and adds to our understanding of the regulation of this critical pathway. However, the study is superficial and somewhat preliminary in parts, and quantitation of the results is largely missing. These shortcomings should be amended before publication can be recommended.

Specific comments

The critical role of PP1 and PP2A mediated dephosphorylation in the activation of CRAF and BRAF kinases has been reported before (Abraham et al., 2000; Dhillon et al., 2002; Jaumot and Hancock, 2001; Strack, 2002), and should be cited and discussed.

The hypothesis that PML-NBs sequester PP1a and thereby restrict ERK activity needs to be further corroborated, e.g. by mapping the binding sites of the PML-PP1a interaction and showing that binding site mutants activate ERK; or showing that a PP1a constitutively targeted to the nucleus via a nuclear localisation signal does not activate ERK.

The role of S6K should be further elaborated. Does knockdown of S6K prevent the activation of ERK by PP1a? Also, as S6K is a downstream target of the ERK pathway it potentially could serve as part of a positive feedback loop to regulate the activation kinetics of ERK? Is this the case?

Does the interaction of PP1a with PML only sequester ERK or does it also affect the PP1a phosphatase activity? This could be tested for instance by measuring PML associated phosphatase activity or by adding purified PML to PP1a phosphatase assays in vitro.

What is the ERK activity in the nuclear fraction plus/minus PML knockdown?

Is 14-3-3g only needed for the cytoplasmic localisation of PP1a or does it also regulate its enzymatic activity?

Figs. 2A and S1A. The activation of ERK by PP1A overexpression is pronounced in HEK293, but rather subtle in LNCaP cells. To obtain a reliable indication of the effect, a quantitation obtained from at least 3 independent experiments should be shown.

Figs. S1C,D. The effect of PP2A is complex, as it can activate RAF by dephosphorylation of inhibitory residues as well as de-activate MEK and ERK by dephosphorylation of activating residues. Thus, measuring ppERK is not sufficient to assess the effects of PP2A. These experiments should be complemented by RAF in vitro kinase assays.

Fig. 2f. The change in ERK activity caused by the different knockdowns is rather small (20% or less in most conditions). These experiments should be quantitated and shown as bar graphs with error bars.

Fig. 4B. The effect of mutating the ERK feedback phosphorylation sites in BRAF is rather subtle. The results should be quantified, and the quantification shown as bar graph with error bars. Given the small effects of mutating the ERK sites, the authors also should consider mutating the other known inhibitory phosphorylation sites in BRAF, i.e. S365, S429, T440 (Guan et al., 2000); S465/467 (Holderfield et al., 2013); S614 (Dernayka et al., 2016)

Minor comments

Fig. 2B. The labelling of the upper panel is incomplete.

Fig. S1E. How much lysate was loaded in the "input" lane compared for what was used for the IP?

Fig. S1G. Tautomycin is not a PP1a specific inhibitor. It inhibits PP1 and PP2 almost with equal potency (Hori et al., 1991).

References

Abraham, D., Podar, K., Pacher, M., Kubicek, M., Welzel, N., Hemmings, B.A., Dilworth, S.M.,

Mischak, H., Kolch, W., and Baccarini, M. (2000). Raf-1-associated protein phosphatase 2A as a positive regulator of kinase activation. *The Journal of biological chemistry* 275, 22300-22304.

Dernayka, L., Rauch, N., Jarbouli, M.A., Zebisch, A., Texier, Y., Horn, N., Romano, D., Gloeckner, C.J., Kriegsheim, A., Ueffing, M., et al. (2016). Autophosphorylation on S614 inhibits the activity and the transforming potential of BRAF. *Cellular signalling* 28, 1432-1439.

Dhillon, A.S., Meikle, S., Yazici, Z., Eulitz, M., and Kolch, W. (2002). Regulation of Raf-1 activation and signalling by dephosphorylation. *The EMBO journal* 21, 64-71.

Guan, K.L., Figueroa, C., Brtva, T.R., Zhu, T., Taylor, J., Barber, T.D., and Vojtek, A.B. (2000). Negative regulation of the serine/threonine kinase B-Raf by Akt. *The Journal of biological chemistry* 275, 27354-27359.

Holderfield, M., Merritt, H., Chan, J., Wallroth, M., Tandeske, L., Zhai, H., Tellew, J., Hardy, S., Hekmat-Nejad, M., Stuart, D.D., et al. (2013). RAF inhibitors activate the MAPK pathway by relieving inhibitory autophosphorylation. *Cancer cell* 23, 594-602.

Hori, M., Magae, J., Han, Y.G., Hartshorne, D.J., and Karaki, H. (1991). A novel protein phosphatase inhibitor, tautomycin. Effect on smooth muscle. *FEBS letters* 285, 145-148.

Jaumot, M., and Hancock, J.F. (2001). Protein phosphatases 1 and 2A promote Raf-1 activation by regulating 14-3-3 interactions. *Oncogene* 20, 3949-3958.

Strack, S. (2002). Overexpression of the protein phosphatase 2A regulatory subunit Bgamma promotes neuronal differentiation by activating the MAP kinase (MAPK) cascade. *The Journal of biological chemistry* 277, 41525-41532.

Reviewer #2 (Remarks to the Author):

Chen and colleagues present biochemical evidence that MAPK pathway may become aberrantly activated in CaP due to PPP1CA (the catalytic subunit of PP1a) genomic amplifications and/or by a non-genomic S6K/PP1a/B-Raf signaling pathway. In the latter scenario, mutations of PML, which normally functions to sequester PP1a into NBs, lead to the accumulation of PP1a in the cytoplasm to dephosphorylate several inhibitory phosphor-sites on B-Raf kinase resulting in MAPK activation. This is an interesting and strong biochemistry study that sheds novel light on how the MAPK pathway, in the absence of component mutations, might become hyper-activated in cancer cells especially CaP cells.

1. One of the gaps in our knowledge from this study is on biology. For example, since PPP1CA is amplified in 17-25% mCRPC, the overexpressed PP1a may promote CaP cell invasion. And if this effect is mediated, at least in part, via activation of B-Raf kinase and downstream MAPK, the pathway inhibitors should blunt PP1a-promoted CaP invasion.
2. In the scenario of increased PP1a activity due to co-deletion of PTEN/PML (without PPP1CA amplification), some quantitative information may help readers appreciate the significance of the proposed signaling pathway. That is, in normal cells (with normal levels of PTEN/PML), how much of PP1a is in the cytoplasm versus sequestered in the PML NBs? How is the protein redistributed in the two cellular compartments in the absence of PML?
3. For data in Figure 2f: Ideally, the authors should repeat the experiment several times and present quantitative data in a bar graph for the changes in p-ERK.
4. How did acute loss of PTEN lead to upregulation of PML?
5. When referring to 'the phosphatase-inactive PP1a mutant (H248K) (page 6), a reference should be provided.

Rebuttal to the Reviewers

We thank both of the Reviewers for their constructive and positive comments that have helped significantly improve the quality, structure and content of this manuscript.

We are pleased that we were able to address each specific comment in full, as outlined in the point-by-point rebuttal section below. Importantly, based on the Reviewers comments and suggestions, we have tremendously improved our study by performing multiple new experiments with both quantitative and qualitative data, as presented in this new version of the manuscript.

We hope that our revised manuscript will now meet the satisfaction of the Reviewers and be deemed suitable for publication in *Nature Communications*.

A detailed rebuttal to each Reviewer's specific comments is included below.

Referee #1 (Remarks to the Author):

In this paper the authors report the positive regulation of ERK signalling by PP1a. Investigating the mechanism they find that PP1a is sequestered in nuclear PML bodies, from where it can be released by S6K mediated phosphorylation and binding to 14-3-3g. It then can translocate to the cytosol to activate BRAF by dephosphorylation of inhibitory residues. This is an interesting finding. Although the regulation of RAF kinases by PP1 and PP2 has been reported before (Abraham et al., 2000; Dhillon et al., 2002; Jaumot and Hancock, 2001; Strack, 2002), the aspect of the spatial regulation is new and adds to our understanding of the regulation of this critical pathway. However, the study is superficial and somewhat preliminary in parts, and quantitation of the results is largely missing. These shortcomings should be amended before publication can be recommended.

We thank the Reviewer for recognizing the novelty of our study and for his/her constructive comments regarding our manuscript. In addressing each of the Reviewers concerns, we have been able to bring additional and more insightful data to the manuscript. We are confident that the Reviewer will appreciate the fact that our conclusion has now been significantly strengthened by these additional quantitative and qualitative data.

Specific comments

The critical role of PP1 and PP2A mediated dephosphorylation in the activation of CRAF and BRAF kinases has been reported before (Abraham et al., 2000; Dhillon et al., 2002; Jaumot and Hancock, 2001; Strack, 2002), and should be cited and discussed.

We apologize for this oversight. We have now included and discussed these references suggested by the Reviewer. They can now be found on **Page 11** of our revised manuscript (**Line #9**).

The hypothesis that PML-NBs sequester PP1a and thereby restrict ERK activity needs to be further corroborated, e.g. by mapping the binding sites of the PML-PP1a interaction and showing that binding site mutants activate ERK; or showing that a PP1a constitutively targeted to the nucleus via a nuclear localisation signal does not activate ERK.

We thank the Reviewer for this comment. As suggested by the Reviewer, we have generated a PP1 α mutant constitutively targeted to the nucleus by fusing the NLS sequence derived from c-Myc (Ray *et al.* Bioconjug Chem 2015, **Fig. 1a**) to C-terminus of PP1 α . Indeed, compared to wild type PP1 α , the NLS-PP1 α displayed a drastically decreased capacity to activate ERK. These results further corroborate the role of subcellular compartmentalization in PP1 α -induced ERK activation and are now shown in our new **Fig. 2e**.

The role of S6K should be further elaborated. Does knockdown of S6K prevent the activation of ERK by PP1 α ?

We have further investigated the role of S6K in PP1 α -induced ERK activation and have found that knockdown of *S6K1* via small interfering RNA largely suppresses PP1 α -induced ERK activation. This again confirms that the activation of ERK by PP1 α is dependent on S6K. These data are now shown in our new **Fig. 3e**.

Also, as S6K is a downstream target of the ERK pathway it potentially could serve as part of a positive feedback loop to regulate the activation kinetics of ERK? Is this the case?

The Reviewer raises an excellent point that S6K-PP1 α -B-Raf-ERK could form a feed-forward loop supporting sustained ERK activation. Based on our current and previous studies (Ma *et al.* Cell 2005), this could be true in the context of *PML* loss or *PPP1CA* amplification. However, we and others have previously also shown that AKT/mTOR/S6K activation triggers a negative feedback on MAPK signaling pathway, presumably as a result of the upstream IRS inactivation induced by S6K (Shah *et al.* Curr Biol 2004; Carracedo *et al.* J Clin Invest 2008) (see also our **Supplementary Fig. 2**). Thus, depending on the genetic context, S6K is a double-edge sword in the regulation of MAPK signaling, since it can act as both a suppressor of ERK activation, in the context of intact *PML* function, and as a promoter of sustained ERK activation, in the context of *PML* loss or *PPP1CA* amplification. We have now also included this important discussion on **Page 12** of our revised manuscript (**Line #2**).

Does the interaction of PP1 α with PML only sequester ERK or does it also affect the PP1 α phosphatase activity? This could be tested for instance by measuring PML associated phosphatase activity or by adding purified PML to PP1 α phosphatase assays in vitro.

We thank the Reviewer for this insightful comment. To determine whether *PML* affects the PP1 α phosphatase activity, we have now performed *in vitro* PP1 α phosphatase assays towards its nuclear targets. The transcription factor cAMP-responsive element binding protein (CREB) is a PP1 α target in the nucleus, where PP1 dephosphorylates CREB at Ser133 and inhibits CREB-mediated transcriptional activation (Hagiwara *et al.* Cell 1992). We have found that purified GST-*PML* does not affect the CREB dephosphorylation exerted by PP1 α *in vitro*, suggesting that *PML* primarily functions as a scaffold/chaperone for PP1 α rather than as a direct regulator of the PP1 α phosphatase activity. These data are now shown in our new **Supplementary Fig. 1m**.

What is the ERK activity in the nuclear fraction plus/minus PML knockdown?

To address the Reviewer's comment, we have examined the transcriptional activity of the Elk-1 transcription factor, a well-established ERK nuclear target (Marais *et al.* Cell 1993), in the absence and in the presence of *PML* knockdown. We have found that knockdown of *PML* results in the activation of the Elk reporter to a greater extent than the siRNA control (**Fig. 1 for the Reviewer**). This is consistent with our complementary studies demonstrating that *PML* loss induces ERK

activation (Chen and Pandolfi *et al.* Nat Genet 2017, manuscript in revision).

Is 14-3-3 γ only needed for the cytoplasmic localisation of PP1 α or does it also regulate its enzymatic activity?

We thank the Reviewer for this critical comment. To determine whether 14-3-3 γ regulates the PP1 α phosphatase activity, we have performed *in vitro* phosphatase assays and have found that 14-3-3 γ has no effect on the PP1 α phosphatase activity towards dephosphorylating B-Raf kinase. These results are now shown in our new **Supplementary Fig. 1n**.

Figs. 2A and S1A. The activation of ERK by PP1A overexpression is pronounced in HEK293, but rather subtle in LNCaP cells. To obtain a reliable indication of the effect, a quantitation obtained from at least 3 independent experiments should be shown.

We thank the Reviewer for this comment. Given that LNCaP cell line is one of the hard-to-transfect cell lines (Fronsdal *et al.* Prostate 2000), the main reason for the observed difference in the activation of ERK by PP1 α between 293T and LNCaP cells, we believe, is the lower transfection efficiency of PP1 α in LNCaP cells. As suggested by the Reviewer, we have now performed three independent transfection experiments using a better transfection reagent, *TransIT-X2* from Mirus Bio LLC, rather than lipofectamine 2000. Along with improved transfection in LNCaP, PP1 α is now able to activate ERK more than six-fold compared to the empty vector control. The representative results, plus quantitation data, are now shown in a revised **Supplementary Fig. 1a**.

Figs. S1C,D. The effect of PP2A is complex, as it can activate RAF by dephosphorylation of inhibitory residues as well as de-activate MEK and ERK by dephosphorylation of activating residues. Thus, measuring ppERK is not sufficient to assess the effects of PP2A. These experiments should be complemented by RAF in vitro kinase assays.

The Reviewer's comments here are well taken. As suggested by the Reviewer, we have now performed *in vitro* kinase assays to show the effect of PP2A on Raf kinase activity. Indeed, PP2A positively regulates B-Raf kinase activity towards phosphorylating MEK. Interestingly, we have also found that PP1 α is a more potent activator of B-Raf than PP2A. These data are now shown in our new **Supplementary Fig. 1b**. Additionally, according to the Reviewer's comments, we have now included a statement to highlight the complex role of PP2A in the regulation of the MAPK signaling. They can now be found on **Page 5** of our revised manuscript (**Last line**).

Fig. 2f. The change in ERK activity caused by the different knockdowns is rather small (20% or less in most conditions). These experiments should be quantitated and shown as bar graphs with error bars.

We agree with the Reviewer and, thus, have now repeated the experiment three times. The representative results, plus quantitation data in a bar graph, are now included in a revised **Fig. 2f** that now appears as **Fig. 2g**.

Fig. 4B. The effect of mutating the ERK feedback phosphorylation sites in BRAF is rather subtle. The results should be quantified, and the quantification shown as bar graph with error bars.

We agree with the Reviewer and, thus, have now repeated the experiment three times. The representative results, plus quantitation data in a bar graph, are now included in a revised **Fig. 4b**.

*Given the small effects of mutating the ERK sites, the authors also should consider mutating the other known inhibitory phosphorylation sites in BRAF, i.e. S365, S429, T440 (Guan *et al.*, 2000); S465/467 (Holderfield *et al.*, 2013); S614 (Dernayka *et al.*, 2016).*

Following the insightful comments from the Reviewer, we have now investigated the effect of PP1 α on other known inhibitory phosphorylation sites suggested by the Reviewer. We have confirmed that the B-Raf mutants devoid of these inhibitory sites indeed exhibit an enhanced ability to activate MAPK signaling. However, they are all still sensitive to PP1 α activation. These results further corroborate that PP1 α appears to exert its effect on B-Raf primarily through the ERK-regulated inhibitory sites and are now shown in our new **Supplementary Fig. 1i-k**.

Minor comments

Fig. 2B. The labelling of the upper panel is incomplete.

We thank the Reviewer for pointing this out. The missing labels in the upper panel of **Fig. 2b** are now included.

Fig. S1E. How much lysate was loaded in the “input” lane compared for what was used for the IP?

“Input” lane contains 10% of total cell extracts used for immunoprecipitation. We have now included this important information in the corresponding figure legend.

Fig. S1G. Tautomycin is not a PP1 α specific inhibitor. It inhibits PP1 and PP2 almost with equal potency (Hori et al., 1991).

The Reviewer is correct that tautomycin is not a PP1 α specific inhibitor. We apologize for this oversight. Given that tautomycin does display a slight preference for PP1 inhibition relative to PP2A inhibition in multiple studies (MacKintosh *et al.* FEBS Lett 1990, **Fig. 2**; Favre *et al.* J Biol Chem 1997, **Fig. 7a**; Swingle *et al.* Methods Mol Biol 2007, **Table 1**), we have now changed the sentence into: “In line with this, PC3 cells treated with tautomycin, **a more selective inhibitor for PP1 α** , displayed a dose-dependent inhibition of EGF-induced ERK phosphorylation”. We have also cited the references mentioned in this comment by the Reviewer and ourselves.

References

- Abraham, D., Podar, K., Pacher, M., Kubicek, M., Welzel, N., Hemmings, B.A., Dilworth, S.M., Mischak, H., Kolch, W., and Baccarini, M. (2000). Raf-1-associated protein phosphatase 2A as a positive regulator of kinase activation. *The Journal of biological chemistry* 275, 22300-22304.
- Dernayka, L., Rauch, N., Jarbouï, M.A., Zebisch, A., Texier, Y., Horn, N., Romano, D., Gloeckner, C.J., Kriegsheim, A., Ueffing, M., et al. (2016). Autophosphorylation on S614 inhibits the activity and the transforming potential of BRAF. *Cellular signalling* 28, 1432-1439.
- Dhillon, A.S., Meikle, S., Yazici, Z., Eulitz, M., and Kolch, W. (2002). Regulation of Raf-1 activation and signalling by dephosphorylation. *The EMBO journal* 21, 64-71.
- Guan, K.L., Figueroa, C., Brtva, T.R., Zhu, T., Taylor, J., Barber, T.D., and Vojtek, A.B. (2000). Negative regulation of the serine/threonine kinase B-Raf by Akt. *The Journal of biological chemistry* 275, 27354-27359.
- Holderfield, M., Merritt, H., Chan, J., Wallroth, M., Tandeske, L., Zhai, H., Tellew, J., Hardy, S., Hekmat-Nejad, M., Stuart, D.D., et al. (2013). RAF inhibitors activate the MAPK pathway by relieving inhibitory autophosphorylation. *Cancer cell* 23, 594-602.
- Hori, M., Magae, J., Han, Y.G., Hartshorne, D.J., and Karaki, H. (1991). A novel protein phosphatase inhibitor, tautomycin. Effect on smooth muscle. *FEBS letters* 285, 145-148.
- Jaumot, M., and Hancock, J.F. (2001). Protein phosphatases 1 and 2A promote Raf-1 activation by regulating 14-3-3 interactions. *Oncogene* 20, 3949-3958.
- Strack, S. (2002). Overexpression of the protein phosphatase 2A regulatory subunit Bgamma promotes neuronal

differentiation by activating the MAP kinase (MAPK) cascade. The Journal of biological chemistry 277, 41525-41532.

References

Ray M, Tang R, Jiang Z, Rotello VM. Quantitative tracking of protein trafficking to the nucleus using cytosolic protein delivery by nanoparticle-stabilized nanocapsules. *Bioconjugate chemistry* **26**, 1004-1007 (2015).

Shah OJ, Wang Z, Hunter T. Inappropriate activation of the TSC/Rheb/mTOR/S6K cassette induces IRS1/2 depletion, insulin resistance, and cell survival deficiencies. *Current biology : CB* **14**, 1650-1656 (2004).

Carracedo A, *et al.* Inhibition of mTORC1 leads to MAPK pathway activation through a PI3K-dependent feedback loop in human cancer. *The Journal of clinical investigation* **118**, 3065-3074 (2008).

Hagiwara M, *et al.* Transcriptional attenuation following cAMP induction requires PP-1-mediated dephosphorylation of CREB. *Cell* **70**, 105-113 (1992).

Marais R, Wynne J, Treisman R. The SRF accessory protein Elk-1 contains a growth factor-regulated transcriptional activation domain. *Cell* **73**, 381-393 (1993).

Chen CH, *et al.* Bidirectional signals transduced by DAPK-ERK interaction promote the apoptotic effect of DAPK. *The EMBO journal* **24**, 294-304 (2005).

Chen M, *et al.* An aberrant SREBP-dependent lipogenic program promotes metastatic cancer. *Nature genetics* Manuscript in Revision (2017).

Fronsdal K, Engedal N, Saatcioglu F. Efficient DNA-mediated gene transfer into prostate cancer cell line LNCaP. *The Prostate* **43**, 111-117 (2000).

MacKintosh C, Klumpp S. Tautomycin from the bacterium *Streptomyces verticillatus*. Another potent and specific inhibitor of protein phosphatases 1 and 2A. *FEBS letters* **277**, 137-140 (1990).

Favre B, Turowski P, Hemmings BA. Differential inhibition and posttranslational modification of protein phosphatase 1 and 2A in MCF7 cells treated with calyculin-A, okadaic acid, and tautomycin. *The Journal of biological chemistry* **272**, 13856-13863 (1997).

Swingle M, Ni L, Honkanen RE. Small-molecule inhibitors of ser/thr protein phosphatases: specificity, use and common forms of abuse. *Methods in molecular biology (Clifton, NJ)* **365**, 23-38 (2007).

Referee #2 (Remarks to the Author):

Chen and colleagues present biochemical evidence that MAPK pathway may become aberrantly activated in CaP due to PPP1CA (the catalytic subunit of PP1 α) genomic amplifications and/or by a non-genomic S6K/PP1 α /B-Raf signaling pathway. In the latter scenario, mutations of PML, which normally functions to sequester PP1 α into NBs, lead to the accumulation of PP1 α in the cytoplasm to dephosphorylate several inhibitory phosphor-sites on B-Raf kinase resulting in MAPK activation. This is an interesting and strong biochemistry study that sheds novel light on how the MAPK pathway, in the absence of component mutations, might become hyper-activated in cancer cells especially CaP cells.

We are extremely pleased that the Reviewer regards our study as interesting as well as strong, and we thank the Reviewer for his/her encouraging review. We also would like to thank the Reviewer for the constructive criticisms that he/she has outlined below.

1. One of the gaps in our knowledge from this study is on biology. For example, since PPP1CA is amplified in 17-25% mCRPC, the overexpressed PP1 α may promote CaP cell invasion. And if this effect is mediated, at least in part, via activation of B-Raf kinase and downstream MAPK, the pathway inhibitors should blunt PP1 α -promoted CaP invasion.

We thank the Reviewer for this critical comment. To address these points, we have established CaP cell lines, stably overexpressing PP1 α , and have used them for cell migration and invasion assays in the absence and in the presence of the MEK inhibitor, U0126. We have found that CaP cells stably overexpressing PP1 α exhibit higher ERK activation along with significantly increased cell migration and invasion. Notably, treatment with the MEK inhibitor, U0126, in CaP cells represses not only basal but also PP1 α -induced cell migration and invasion. These functional data, together with the human genetic and mechanistic analyses, implicate *PPP1CA* as a pro-metastatic proto-oncogene in human CaP and MAPK signaling as one of the key downstream effectors of PP1 α -induced cell invasiveness. These important new results are now shown in our new **Fig. 4e** and **Supplementary Fig. 11**.

2. In the scenario of increased PP1 α activity due to co-deletion of PTEN/PML (without PPP1CA amplification), some quantitative information may help readers appreciate the significance of the proposed signaling pathway. That is, in normal cells (with normal levels of PTEN/PML), how much of PP1 α is in the cytoplasm versus sequestered in the PML NBs? How is the protein redistributed in the two cellular compartments in the absence of PML?

We thank the Reviewer for this great suggestion. To quantify the changes in the cellular distribution of PP1 α upon co-loss of *PTEN/PML*, we have now repeated the WI-38 cellular fractionation assays in **Fig. 2d** three times. WI-38 cells are normal human diploid fibroblasts with intact PTEN and PML protein expression. We have found that PP1 α primarily localizes in the nucleus (80% nuclear and 20% cytoplasmic) in WI-38 control cells, whereas the cellular distribution of PP1 α was reversed upon knockdown of *PTEN* and *PML* (30% nuclear and 70% cytoplasmic). The representative results, plus quantitation data in a bar graph, are now included in a revised **Fig. 2d**.

3. For data in Figure 2f: Ideally, the authors should repeat the experiment several times and present quantitative data in a bar graph for the changes in p-ERK.

We agree with the Reviewer and, thus, have now repeated the experiment three times. The representative results, plus quantitation data in a bar graph, are now included in a revised **Fig. 2f** that now appears as **Fig. 2g**.

4. How did acute loss of *PTEN* lead to upregulation of PML?

We thank the Reviewer for bringing up this question. We have previously shown that acute loss of *PTEN* results in not only AKT/mTOR activation, but also consistent induction of p53 (Chen *et al.* Nature 2005, **Fig. 2c**). PML is a direct p53 target (de Stanchina *et al.* Mol Cell 2004, **Fig. 5**), while mTOR activation can enhance PML protein synthesis even in the absence of p53 (Scaglioni *et al.* EMBO Mol Med 2012, **Fig. 3a**). Therefore, both transcriptional and post-transcriptional mechanisms contribute to upregulation of PML upon *PTEN* loss. We have now also included this important information on **Page 6** of our revised manuscript (**Line #7**).

5. When referring to 'the phosphatase-inactive PP1a mutant (H248K) (page 6), a reference should be provided.

We thank the Reviewer for pointing this out. We have now cited the reference where the PP1 α H248K mutant was originally studied and reported (Zhang *et al.* Biochemistry 1996) in our revised manuscript.

References

Chen Z, *et al.* Crucial role of p53-dependent cellular senescence in suppression of Pten-deficient tumorigenesis. *Nature* **436**, 725-730 (2005).

de Stanchina E, *et al.* PML is a direct p53 target that modulates p53 effector functions. *Molecular cell* **13**, 523-535 (2004).

Scaglioni PP, *et al.* Translation-dependent mechanisms lead to PML upregulation and mediate oncogenic K-RAS-induced cellular senescence. *EMBO molecular medicine* **4**, 594-602 (2012).

Zhang J, Zhang Z, Brew K, Lee EY. Mutational analysis of the catalytic subunit of muscle protein phosphatase-1. *Biochemistry* **35**, 6276-6282 (1996).

REVIEWERS' COMMENTS:

Reviewer #1 (Remarks to the Author):

The authors have invested a lot of work to perform additional experiments, which have substantially improved the paper. The work makes an important contribution to our understanding of the crossregulation between the PML and RAF pathways and is ideally suited for publication in Nature Communications.

Reviewer #2 (Remarks to the Author):

The authors have conscientiously and adequately addressed my questions and the revised manuscript has been significantly strengthened.

Manuscript # NCOMMS-16-28227-A

Referee #1 (Remarks to the Author):

The authors have invested a lot of work to perform additional experiments, which have substantially improved the paper. The work makes an important contribution to our understanding of the crossregulation between the PML and RAF pathways and is ideally suited for publication in Nature Communications.

We thank the Reviewer for his/her positive assessment of our work and are pleased that he/she feels our manuscript is ready for publication.

Referee #2 (Remarks to the Author):

The authors have conscientiously and adequately addressed my questions and the revised manuscript has been significantly strengthened.

We thank the Reviewer for his/her positive assessment of our work and are pleased that he/she feels our manuscript is significantly strengthened and ready for publication.